# Phenotypic, Chemotaxonomic, and Genome-Based Classification of *Phyllobacterium* Strains: Two Proposed Novel Species, *Phyllobacterium chamaecytisi* sp. nov. and *Phyllobacterium lublinensis* sp. nov

**DOI:** 10.3390/biology14081024

**Published:** 2025-08-08

**Authors:** Sylwia Wdowiak-Wróbel, Karolina Włodarczyk-Ciekańska, Monika Marek-Kozaczuk, Marta Palusińska-Szysz, Piotr Koper, Jerzy Wielbo

**Affiliations:** 1Department of Genetics and Microbiology, Maria Curie-Sklodowska University, Akademicka 19 Str., 20-033 Lublin, Poland; sylwia.wdowiak-wrobel@mail.umcs.pl (S.W.-W.); monika.marek-kozaczuk@mail.umcs.pl (M.M.-K.); marta.palusinska-szysz@mail.umcs.pl (M.P.-S.); piotr.koper@mail.umcs.pl (P.K.); 2Independent Laboratory of Cancer Diagnostics and Immunology, Faculty of Medicine, Medical University of Lublin, Chodźki 1 (Collegium Universum), 20-093 Lublin, Poland; karolina.wlodarczyk-ciekanska@umlub.pl

**Keywords:** *Phyllobacterium*, *Chamaecytius albus*, taxonomy, endophytes

## Abstract

Plants can be colonized by numerous and diverse bacteria. Studies of microorganisms isolated from the roots of *Chamaecytisus albus* (Spanish broom) plants identified two bacterial strains with characteristics different from those previously identified. Detailed genetic and morphological studies as well as analysis of selected organic factors indicate that these strains are related to known *Phyllobacterium* species, but differ sufficiently from them to be considered distinct species. For these two new species, we propose the names *Phyllobacterium chamaecytisi* (from *Chamaecytisus*, the host plant from which strains were isolated) and *Phyllobacterium lublinensis* (referring to Lublin, the region in eastern Poland where the host plant was collected). Interestingly, although these strains do not belong to the group of rhizobia, i.e., bacteria establishing symbiosis with legumes, which also includes *Chamaecytisus*, they were found in root nodules - the part of the plant colonized by symbiotic bacteria. The research conducted has expanded our knowledge of the species richness of bacteria from the genus *Phyllobacterium* and the biodiversity of microbial communities inhabiting plant tissues.

## 1. Introduction

Bacteria belonging to the genus *Phyllobacterium* can be commonly found on the rhizoplane and phylloplane of plants, as well as inside plant tissues. The genus *Phyllobacterium* was originally described by Knösel [1,2,3], and the first well-characterized species was *P. myrsinacearum* originating from *Ardisia* leaf nodules [4].

Until now, fourteen recognized *Phyllobacterium* species have been described. Numerous type strains were isolated from root nodules of different legumes like *Trifolium pratense* (*P. trifolii*) [5], *Lathyrus numidicus* (*P. ifriquiense*) [6], *Astragalus algerianus* (*P. leguminum*) [6], *Phaseolus vulgaris* (*P. endophyticus*) [7], *Lotus corniculatus* (*P. loti*) [8], *Sophora florescens* (*P. sophorae*) [9], *Lotus lancerottensis* (*P. salinisoli*) [10], or *Oxytropis triphylla* (*P. zundukense*) [11]. However, the capacity of these isolates to induce nodulation was demonstrated only for *P. trifolii* and *P. sophorae* [5,9] suggesting that most of mentioned bacteria are non-symbiotic plant endophytes. Some *Phyllobacterium* species like *P. bourgognense* or *P. brassicacearum* were found inside tissues of non-leguminous plants; other were isolated from soil (*P. phragmitis, P. pellucidum*) [12,13] or from volcanic rock (*P. catacumbae*) [14]. Moreover, different *Phyllobacterium* strains have also been found in water [15] or associated with unicellular Eukaryotes [16,17], which shows the great variety of habitats which can be colonized by these microorganisms, and the ability of phyllobacteria to adaptation to free-living, associative, and even endosymbiotic lifestyles.

*Chamaecytisus albus* (white Spanish broom) is a legume shrub that can be found in only one natural habitat, near Hrubieszow in south-eastern Poland. The Polish natural habitat of *C. albus* is still shrinking, and so is the number of living plants in this environment; therefore, this species was described as “critically endangered” in 2016 in the “Polish Red List of Fern and Flowering Plants” [18]. Bacterial endophytes of these plants were recently isolated and described [19].

The aim of the current study was to demonstrate the distinct taxonomic position of the *Phyllobacterium*-related isolates KW56 and 2063, based on 16S rRNA gene sequencing, whole-genome sequence analysis, the comparison of morphological and physiological features, and comparative whole-cell fatty acid analysis.

## 2. Materials and Methods

### 2.1. Strains Isolation

Strains were isolated from root nodules of *Chamaecytisus albus* (a plant growing near Hrubieszow in the South-Eastern region of Poland, 50°48′09″ N, 23°53′31″ E). Nodules were harvested and surface-sterilized; they were rinsed several times with sterile water, incubated with 0.1% HgCl_2_ for 5 min, rinsed several times with sterile water again, incubated with 70% ethanol for 5 min, and finally rinsed several times with sterile water. The suspension was made by crushing single nodules in sterile water. It was transfered on yeast–mannitol medium (YEM) and incubated at 28 °C for 7 days [20]. Bacteria isolated from *Chamaecytisus albus* root nodules were purified by streaking several times on YEM agar. The purified strains were kept in YEM medium at 4 °C as well as at −70 °C in YEM medium with 15% (*v*/*v*) glycerol.

### 2.2. Whole-Genome Sequencing

DNA was extracted from overnight cultures using a Genomic Mini AX Bacteria Spin Kit (A&A Biotechnology, Gdansk, Poland). Extractions were performed as described in the manufacturer’s protocol for Gram-negative bacteria. Genomic DNA was sequenced via StarSEQ GmbH, Mainz, Germany.

Illumina raw pair-end 150 bp reads were quality controlled and trimmed via FastQC v0.11.9 (https://github.com/s-andrews/FastQC, accessed on 12 October 2023), FastQ Screen v0.14.1 (https://www.bioinformatics.babraham.ac.uk/projects/fastq_screen, accessed on 12 October 2023), and Trimmomatic [21], using standard parameters. Filtered reads were assembled into contigs using Spades (3.13.0) [22]. The draft genomes assemblies were deposited at GenBank under accession numbers GCA_020164435.1 and GCA_020164455.1.

### 2.3. Genome Annotation

The genome was annotated by the NCBI Prokaryotic Genome Annotation Pipeline (PGAP) [23]. For comparison, the genome was also submitted to the Comprehensive Genome Analysis Service at PATRIC (www.patricbrc.org) [24] and annotated using RAST tool kit v1.073 [25]. The genome was checked for completeness and contamination using EvalG and EvalCon [26]. The genome maps were prepared using the circos v0.69-9 tool as part of the Comprehensive Genome Analysis service on PATRIC [24].

### 2.4. Phylogenomic Analyses

Single-gene phylogenies based on 16S rRNA gene sequences were inferred by the GGDC web server [27] available at http://ggdc.dsmz.de/ using the DSMZ phylogenomics pipeline [28] adapted to single genes. A multiple sequence alignment was created with MUSCLE v3.8.31 [29]. Maximum likelihood (ML) and maximum parsimony (MP) trees were inferred from the alignment with RAxML v8 [30] and TNT [31], respectively. For ML, rapid bootstrapping in conjunction with the autoMRE bootstrapping criterion [32] and subsequent search for the best tree was used; for MP, 1000 bootstrapping replicates were used in conjunction with tree-bisection-and-reconnection branch swapping and ten random sequence addition replicates. The sequences were checked for a compositional bias using the Χ^2^ test as implemented in PAUP* v4.0 Beta [33].

Whole-genome phylogenomic analysis utilizing digital DNA-DNA hybridization (dDDH) was calculated using the Type Strain Genome Server (TYGS) available at https://tygs.dsmz.de/ [34] based on the available genomes of closely related neighbors. For comparison and confirmation, the ANI analysis was also performed using the JSpeciesWS webserver [35] using the same set of genomes. Default parameters were used for all software unless otherwise specified.

### 2.5. Phenotypic Analyses

The pure cultures of *Chamaecytisus albus* isolates KW56 and 2063 were phenotypically characterized by the Biolog GEN III system (Biolog Inc., Hayward, CA, USA), following the manufacturer’s instructions. The GEN III MicroPlates™ (Biolog Inc., Hayward, CA, USA) enables micro-testing of bacteria, assessing the ability to metabolize 71 carbon sources and containing 23 chemical sensitivity assays. The GEN III plates contained tetrazolium redox dye, which was used to colorimetrically indicate positive reactions. Bacterial colonies were transferred to inoculating fluid A (IFA) to generate bacterial cell suspensions, the transmittance of which was adjusted to achieve a 95% using a turbidimeter (Laxco Inc., Mill Creek, WA, USA). 100 µL of the cell suspension was dispensed into each well. The color development was monitored every 24 h as absorbance with Synergy™ H1 microplate reader (Agilent Technologies, Santa Clara, CA, USA) at a wavelength of 590 and 750 nm over three days. The most consistent readings came from three-day-old Biolog plates, and these data were used in the analyses. Enzymatic characterization of bacterial isolates was performed using API ZYM strips (BioMérieux, Marcy-l’Étoile, France) according to the manufacturer’s protocol.

### 2.6. Preparation of Fatty Acid Methyl Esters (FAME)

The strains were cultivated for 48 h in a liquid TY medium (tryptone 5.0 g, yeast extract 3.0 g, CaCl_2_ × 2 H_2_O 0.9 g, distilled water 1000.0 mL, pH 7) at 28 °C with aeration by vigorous shaking. Then, 5 mg of bacterial mass was suspended in 1 mL of 2 M methanolic HCl [(prepared from acetyl chloride, Sigma-Aldrich, St. Louis, MO, USA)]. The samples were heated at 85 °C for 18 h. After removing the excess reagent on the evaporator (Büchi Labortechnik AG, Flawil, Switzerland), FAMEs were extracted with a mixture of chloroform water (1:2 *v*/*v*). Extraction of FAMEs was repeated twice, and the pooled chloroform phases dehydrated on columns with anhydrous sodium sulfate were dried in a nitrogen stream. FAMEs were converted into trimethylsilyl (TMS) ether derivatives by adding 30 µL TMS reagent [HMDS/TMCS/pyridine (3:1:9, *v*/*v*/*v*), Sigma-Aldrich, St. Louis, MO, USA]. The FAME samples were analyzed by GC/MS using an Agilent 7890A-5975C instrument equipped with an HP-5MS capillary column (30 m × 0.25 mm; Supelco, St. Louis, MO, USA). Helium was used as carrier gas and the temperature program was initially 150 °C for 5 min, then raised to 310 °C at a ramp rate of 3 °C/min, final time of 20 min.

FAMEs were identified mainly based on their chromatographic and mass spectral characteristics, as well as comparison of their retention times with those of authentic standards. The relative content (%) of each fatty acid was calculated from the ratio of the area of its peak to the total area of all peaks. FAMEs of the studied strains were prepared in three independent experiments.

## 3. Results

### 3.1. The Origination of Strains

Strains KW56 and 2063 were obtained from root nodules of *Chamaecytisus albus* growing in Hrubieszów, Poland, during the study of rhizobia and other nodule endophytes of this shrub, in May 2019. Phenotypic traits of these strains were different; however, none of them was able to nodulate *C. albus* plants [19].

### 3.2. Genetic Analyses

Genetic analyses included traditional 16S rRNA gene sequences comparison as well as an overall genome-related index (OGRI) using parameters like dDDH (digital DNA-DNA hybridization) and ANI (average nucleotide identity) which are considered appropriate tools for delineating new species [36].

The comparison of 16S rDNA sequences of studied strains and *Phyllobacterium* type strains revealed that KW56^T^ and 2063^T^ have different closest relatives (Figure 1); *Phyllobacterium* strain KW 56^T^ was located in one clade together with *P. loti* S 658^T^ and *P. trifolii* CECT 7015^T^, whereas *Phyllobacterium* strain 2063^T^ was located in a clade containing also *P. sophorae* CCBAU 03422^T^, *P. brassicacearum* STM 196^T^, and *P. brassicacearum* 29-15.

For the 2063^T^ strain, 16S rRNA gene sequence similarity varied from 96.7% (*P. catacumbae* CSC 19^T^) to 99.6% (*P. brassicacearum* 29-15 and *P. brassicacearum* STM 196^T^). These values calculated for the KW56 strain ranged from 96.1% (*P. leguminum* ORS 1419^T^) to 100% (*P. trifolii* CECT 7015^T^ and *P. loti* S 658^T^) (Table 1).

Features of the KW56^T^ and 2063^T^ genomes are summarized in Table 2. In total, 93 contigs for KW56^T^ were obtained, with 100-fold coverage. The average contig length was 58,081 bp, with the largest contig being 733,223 bp and the shortest being 6055 bp. The KW56 has a genome of 6.4 Mb (226.4 kb scaffold N50) with 5857 protein-coding genes and 54 RNA genes (3 rRNAs, 47 tRNAs, 4 ncRNAs) (Table 2, Appendix A).

In the case of 2063^T^ strain in total, 45 contigs were obtained, with 100-fold coverage. The average contig length was 187.708 bp, with the largest contig being 598,383 bp and the shortest being 202 bp. The 2063^T^ strain has a genome of 4.5 Mb (188.4 kb scaffold N50) with 4231 protein-coding genes and 52 RNA genes (3 rRNA, 45 tRNAs, 4 ncRNAs) (Table 2, Appendix A).

The DNA G+C content of strains KW56^T^ and 2063^T^ were 56 and 57.5 mol%, respectively. These values are within the range of DNA G+C content reported for members of the genus *Phyllobacterium* (51–58 mol%) [6].

The predicted genes were functionally categorized using the SEED subsystems [34] at the RAST server. The genome of strains KW56^T^ and 2063^T^ were additionally mapped to the seed subsystem to attain high-quality genome annotation via BlastKoala. The distributions of genes linked to subsystems in 11 different categories are shown in Figure 2A,B.

Most of *Phyllobacterium* genomes available in databases are draft genomes; moreover, they differ from each other in size, therefore “formula d4” (i.e., sum of all identities found in high-scoring pairs divided by total genome length), which is independent of genome length, was used in dDDH comparisons. These analyses revealed that dDDH similarity level for KW56^T^ compared with other strains varied from 19.7 (*P. phragmitis* 1N-3^T^) to 58.6% (*P. trifolii* CECT 7015^T^), and this value for 2063^T^ varied from 19.5% (*P. salinisoli* LLAN 61^T^) to 28.3% (*P. brassicacearum* 29-15 and *P. brassicacearum* STM 196^T^) (Table 3).

Studying ANIb values showed that KW56^T^ strain revealed the highest similarity to *P. trifolii* CECT 7015^T^ (93.52%) and the lowest to *P. leguminum* ORS 1419^T^ (71.72%). The same analyses performed for the 2063 strain revealed the highest similarity to *P. brassicacearum* 29-15 (84.08%) and the lowest to *P. salinisoli* LLAN 61^T^ (72.31%) (Table 3).

### 3.3. Phenotypic Analyses

Strain KW56^T^ has the general characteristics of the genus *Phyllobacterium*. It is a Gram-negative, aerobic rod. The optimal growth temperature is 28 °C. Growth on YMA occurs at 37 °C but not at 40 °C. Growth occurs in YM broth with 1, 2, or 3% NaCl and YM broth at pH 6–8, but not at pH 5. Catalase and oxidase activities are positive. Phosphate solubilization, siderophores, and IAA production is positive [19].

Strain KW56^T^ utilizes the following sugars: D-maltose, D-cellobiose, gentiobiose, sucrose, D-turanose, N-acetyl-d-glucosamine, N-acetyl-β-d-mannosamine, N-acetyl-neuraminic acid, α-d-glucose, D-mannose, D-fructose, D-galactose, D-fucose, L-fucose, L-rhamnose. It uses such polyvalent alcohols as D-sorbitol, D-mannitol, D-arbitol, myo-inositol, glycerol. Assimilation of hexose acids such as D-galacturonic acid, L-galactonic acid lactone, D-glucuronic acid, glucuronamide, quinic acid, and D-saccharic acid was positive. Among the carboxylic acids, esters, and fatty acids tested, substrates such as L-lactic acid, D-malic acid, L-malic acid, bromo-succinic acid, tween 40, α-hydroxy-butyric acid, β-hydroxy-D, L-butyric acid, α-keto-butyric acid, acetoacetic acid, propionic acid, acetic acid, and formic acid were used. The strain KW56 also used different amino acids: glycyl-L-proline, L-alanine, L-apartic acid, L-serine.

This strain was sensitive to pH 5, 8% NaCl, and niaproof 4. No positive reaction was observed for aztreonam, sodium butyrate, lithium chloride, potassium tellurite, fusidic acid, troleandomycin, rifamycin SV, minocycline, lincomycin, or guanidine HCl. An API ZYM kit showed positive enzyme activities for alkaline phosphatase, esterase (C4), leucine arylamidase, acid phosphatase, and naphthol-AS-BI-phosphohydrolase, but not for lipase (C14), esterase lipase (C8), valine arylamidase, cystine arylamidase, trypsin, α-glucosidase, α-galactosidase, β-galactosidase, β-glucuronidase, β-glucosidase, *N*-acetyl-β-glucosaminidase, α-mannosidase, or α fucosidase.

Strain 2063^T^ has the general characteristics of the genus *Phyllobacterium*. It is a Gram-negative, aerobic rod. The optimal growth temperature is 28 °C. Growth on YMA occurs at 37 °C but not at 40 °C. Growth occurs in YM broth with 1, 2, or 4% NaCl and YM broth at pH 6–8, but not at pH 5. Catalase and oxidase activities are positive. Siderophores and IAA production is positive [19].

This strain utilizes the following sugars: D-maltose, D-trehalose, D-cellobiose, gentiobiose, D-turanose, D-glucose, D-mannose, D-fructose, D-galactose, D-fucose, L-fucose, N-acetyl-d-glucosamine. It uses such polyvalent alcohols as D-sorbitol, D-mannitol, D-arabitol, myo-inositol, and glycerol. Assimilation of L-glutamic acid, L-histidine, quinic acid and glucuronamide is positive. Among the carboxylic acids, esters and fatty acids tested, L-lactic acid, keto-glutaric acid, acetoacetic acid, propionic acid, acetic acid, and formic acid were used. No positive reaction was observed for fusidic acid, minocycline, or guanidine HCl. The 2063^T^ strain was sensitive to pH 5, 8% NaCl, minocycline, and niaproof 4.

API ZYM kit, showed positive enzyme activities for alkaline phosphatase, esterase (C4), leucine arylamidase, trypsin, acid phosphatase, α-glucosidase, and naphthol-AS-BI-phosphohydrolase, but not for lipase (C14), esterase lipase (C8), valine arylamidase, cystine arylamidase, α-galactosidase, β-galactosidase, β-glucuronidase, β-glucosidase, *N*-acetyl-β-glucosaminidase, α-mannosidase or α fucosidase.

The discriminatory characters for KW56 and 2063 strains, as well as closely related *Phyllobacterium* species *P. brassicacearum* LMG 22836^T^ and *P. trifolii* PETP02^T^ are listed in Table 4.

### 3.4. Fatty Acid Porofiles of KW56^T^ and 2063^T^ Strains

Strain KW56^T^ synthesized 21 different fatty acids with 14 to 21 carbon atoms, including saturated, unsaturated, branched, hydroxy acids as well as FAs with a cyclopropane ring (Table 5). The 16:0 FA was dominant among the saturated acids. Monounsaturated FA comprised 16:1, 17:1, 18:1, 19:1. This strain contained two cyclopropyl fatty acids, of which cyclopropyl 19:0 was the predominant acid. The second most abundant acid was 3-OH 16:0. This strain produced also 2-OH 19:1 and 11-OH 19:0 fatty acids.

Analysis of FAMEs performed for 2063^T^ strain showed the presence of a varied composition including straight-chain saturated and unsaturated, branched-chain, cyclopropyl-ringed, and hydroxy FAs. The dominant components in this strain were cyclopropyl 19:0 (42%) and 18:1 (16%) (Table 3). This strain was characterized by a high content of hydroxy fatty acids, which together accounted for approx. 20%. Among hydroxy FAs, 3-OH was the predominant acid. The fatty acid 16:0 was the most abundant of the saturated FAs of the 2063^T^ strain.

## 4. Discussion

As presented above, the 16S rDNA sequences of the studied strains did not differ from those obtained from some reference strains. However, the whole-genome analysis showed large differences between compared genomes. The threshold proposed for delineation of new species is <70% for dDDH and <95–96% for ANI [36,37], and values obtained for KW56^T^ and 2063^T^ were below this threshold; therefore, these strains can be claimed as new *Phyllobacterium* species according to the obtained OGRI values.

Based on the RAST annotation, the individual predicted genes were assigned to general functional categories. The assignment of individual CDSs to functional categories was illustrated in the genomic maps for both analyzed strains (Online resource 1). For both genomes, the number of genes in each category was similar; however, differences were found. The genes involved in basic metabolism were the largest group (880 for KW56^T^ and 764 for 2063^T^), followed by energy production (337 and 248 respectively) and protein processing (226 and 220).

Differences found between the genomes of KW56^T^, 2063^T^, and their closest relatives (*P. trifolii* and *P. brassicacearum*, respectively) were supported in the physiological and metabolic properties of the strains. the studied strains differed from the reference strains in general physiological traits like growth temperature or pH and salt tolerance, as well as in metabolic potential (e.g., utilization of numerous specific organic acids, aminoacids and other carbon and energy sources).

Comparative analysis of cellular fatty acids showed a similarity of KW56^T^ strain to its closest relative *P. trifolii* CECT 7015^T^ characterized by major acids 16:0, 18:1ω7c, and 19:0 cyclopropyl ω8c (Table 3). The KW56^T^ strain differed from the *P. trifolii* strain in the amount of 19:0 cyclopropyl, which was three times higher in the novel strain, in the amount of 18:1, which was lower [5]. The presence of significant amounts of 3-OH 14:0, branched 19:0, and 21:0 fatty acids clearly distinguished the KW56^T^ strain from *P. trifolii* CECT 7015^T^. These fatty acids may be chemotaxonomic markers of this novel strain.

The predominance of fatty acids such as 16:0 and cyclopropyl 19:0 was similar to that of the closest related *P. brassicacearum* 29-15 strain [8]. The 2063^T^ strain contained a significant amount of 18:1 fatty acid, which was not present in the *P. brassicacearum* 29-15 strain. A unique signature fatty acid, a19:1, was found only in the 2063^T^ strain and represented 3% of the total. This novel strain could be differentiated from *P. brassicacearum* 29-15 also due to the presence of 3-OH 14:0, 3-OH 17:0, 3-OH 18:0.

In conclusion, the phenotypic and genotypic data presented in this study demonstrate that the KW56^T^ and 2063^T^ strains belong to a separate species in the genus *Phyllobacterium*. Therefore, we conclude that the KW56^T^ and 2063^T^ strains are representatives of a new species, for which we propose the name *P. chamaecytisi* KW56^T^, with strain KW56^T^ as the type strain. The 2063^T^ strain is representative of a second new species, for which we propose the name *P. lublinensis* 2063^T^, with strain 2063^T^ as the type strain.

## 5. Conclusions

The bacteria *Phyllobacterium chamaecytisi* KW56^T^ and *Phyllobacterium lublinensis* 2063^T^, isolated from root nodules of *Chamaecytisus albus*, represent novel species within the genus *Phyllobacterium*. Both strains are capable of forming endophytic associations with their leguminous host plant. Genomic analyses, including ANI and dDDH, confirmed their distinction from previously described *Phyllobacterium* species.

*P. chamaecytisi* KW56^T^ and *P. lublinensis* 2063^T^ differ in genome size, GC content, and fatty acid profiles, particularly in the proportions of hydroxy and cyclopropyl fatty acids. Phenotypic characterization revealed distinct metabolic and enzymatic traits, including specific carbon source utilization and enzymatic activities.

Both strains exhibit plant-associated traits and may contribute to plant health and nitrogen metabolism through endophytic colonization. This research expands current knowledge of the genus *Phyllobacterium*, particularly in relation to symbiotic bacteria associated with rare legume species.

The discovery of *P. chamaecytisi* KW56^T^ and *P. lublinensis* 2063^T^ provides new insights into bacterial diversity within root nodule microbiomes and lays the foundation for further studies on plant–microbe interactions and potential agricultural applications.

Protologue description of *Phyllobacterium chamaecytisi* sp. nov. and *Phyllobacterium lublinensis* sp. nov.

*Phyllobacterium chamaecytisi* sp. nov.

*Phyllobacterium chamaecytisi* (*cha.mae.cy.ti’si*. N.L. gen. n. *chamaecytisi*, from *Chamaecytisus*, the host plant from which the type strain was isolated).

*Phyllobacterium chamaecytisi* sp. nov. is represented by strain KW56ᵀ (=DSM 113831ᵀ = GCA_020164455.1), which was isolated in May 2019 from root nodules of *Chamaecytisus albus* collected in Hrubieszów, Poland (50°48′09″ N, 23°53′31″ E). Cells are Gram-negative, non-spore-forming rods. The strain grows aerobically on yeast extract mannitol agar (YMA) at 28 °C and 37 °C but not at 40 °C. It tolerates up to 3% (*w*/*v*) NaCl and grows at pH values between 6.0 and 8.0. The strain is catalase- and oxidase-positive. It does not fix nitrogen, and no *nodABC* genes are present in the genome.

The major fatty acid is cyclopropyl 19:0 (36%), with additional components including 3-OH 14:0, anteiso 19:1, and 21:0, which collectively distinguish the strain from *Phyllobacterium trifolii*. The 16S rRNA gene sequence (NZ_JAIQWW010000048.1) of strain KW56ᵀ is identical (100%) to that of *P. trifolii* CECT7015ᵀ. However, genome-based metrics support its designation as a novel species, with an average nucleotide identity (ANI) of 93.5% and digital DNA–DNA hybridization (dDDH) value of 58.6% compared to *P. trifolii*.

Strain KW56ᵀ differs from *P. trifolii* in several phenotypic traits. It can grow at 4% NaCl and pH 5.0, is urease-positive, and is capable of utilizing L-serine, L-alanine, propionate, acetic acid, and Tween 40 as carbon sources. Biolog GENIII and API ZYM enzymatic profiles clearly differentiate this strain from its closest phylogenetic relative.

*Phyllobacterium lublinensis* sp. nov.

*Phyllobacterium lublinensis* (*lu.bli.nen’sis*. N.L. fem. adj. *lublinensis*, referring to Lublin, the region in eastern Poland where the host plant was collected).

*Phyllobacterium lublinensis* sp. nov. is represented by strain 2063ᵀ (=DSM 113830ᵀ = GCA_020164435.1), which was isolated in May 2019 from root nodules *of Chamaecytisus albus* collected in Hrubieszów, Poland. Cells are Gram-negative, rod-shaped, and non-spore-forming. The strain grows aerobically on YMA at 28 °C and 37 °C, but not at 40 °C. It tolerates up to 4% (*w*/*v*) NaCl and grows in a pH range of 6.0–8.0. The strain is catalase and oxidase-positive. It does not fix nitrogen, and no *nodABC* genes are present in the genome.

The major fatty acids are cyclopropyl 19:0 (42%) and 18:1. Additional key components, including 3-OH 14:0, 3-OH 17:0, and 3-OH 18:0, are present and allow differentiation from *Phyllobacterium brassicacearum*, its closest phylogenetic neighbor. The 16S rRNA gene sequence (NZ_JAIUDQ010000029.1) of strain 2063ᵀ shares 99.6% identity with that of *P. brassicacearum* STM 196^T^. However, ANI (84.1%) and dDDH (28.3%) values are below the species-level thresholds, confirming its status as a novel species.

Strain 2063ᵀ is distinguishable from *P. brassicacearum* by its ability to grow in the presence of 4% NaCl and to metabolize α-ketoglutarate, D-glucuronate, acetic acid, and propionate. It also exhibits trypsin activity. Biolog and API ZYM profiles further support its differentiation from *P. brassicacearum*.

## Figures and Tables

**Figure 1 biology-14-01024-f001:**
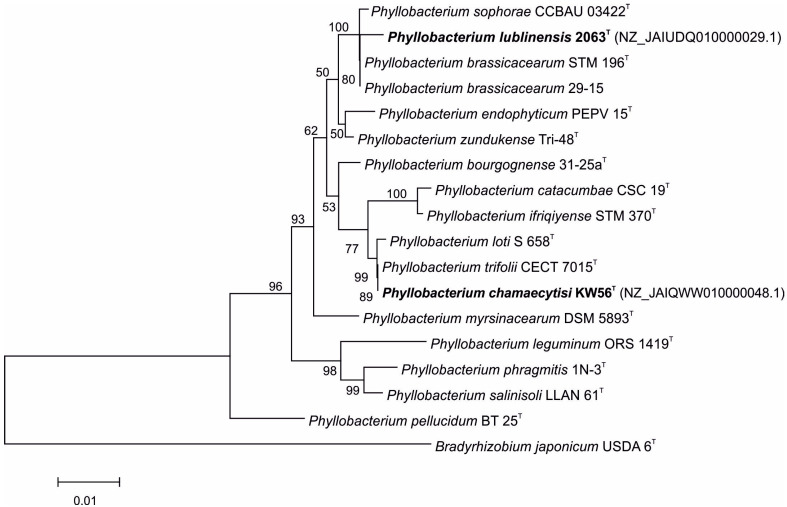
Phylogenetic tree based on 16S rRNA gene sequences of studied and reference *Phyllobacterium* strains.

**Figure 2 biology-14-01024-f002:**
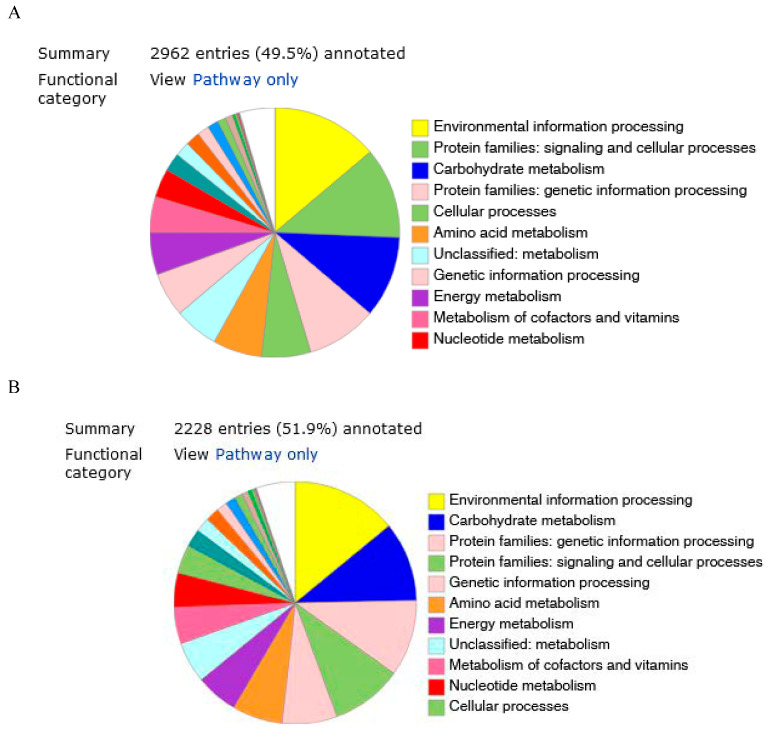
Genome annotation of 2063^T^ (**A**) and KW56^T^ (**B**) by BlastKoala.

**Table 1 biology-14-01024-t001:** 16S rDNA sequence similarity calculated for *Phyllobacterium* reference strains and KW56 as well as 2063 strains.

Strain	16S rRNA Gene Sequence Similarity (%)	
	2063^T^	KW56^T^
*P. bourgognense* 31-25 a^T^	98.5	99.2
*P. bourgognense* STM 201^T^	98.4	99.1
*P. brassicacearum* 29-15	99.6	98.8
*P. brassicacearum* STM 196^T^	99.6	98.7
*P. catacumbae* CSC 19^T^	96.7	98.4
*P. endophyticum* PEPV 15^T^	98.8	98.7
*P. ifiqiyense* STM 370^T^	97.4	99.0
*P. leguminum* ORS 1419^T^	97.1	96.1
*P. loti* S 658^T^	98.4	100.0
*P. myrsinacearum* IAM 13584^T^	98.2	98.3
*P. myrsinacearum* IAM 13587^T^	98.2	98.2
*P. pellucidum* BT 25^T^	97.7	96.9
*P. phragmitis* 1N-3^T^	97.1	96.9
*P. myrsinacearum* DSM 5893^T^	98.3	98.3
*P. salinisoli* LLAN 61^T^	97.2	97.2
*P. sophorae* CCBAU 03422^T^	99.0	98.2
*P. trifolii* CECT 7015^T^	98.4	100.0
*P. trifolii* PETP 02^T^	98.3	99.9
*P. zundukense* Tri-48^T^	98.9	99.0
2063^T^	100.0	98.4
KW56^T^	98.4	100.0

**Table 2 biology-14-01024-t002:** Genome organization.

Genomic Data	*Phyllobacterium chamaecytisi* KW56^T^	*Phyllobacterium lublinensis* 2063^T^
DDBJ/EMBL/GenBank accesion number	GCA_020164455.1	GCA_020164435.1
Sequence size (bp)	6.4 Mb	4.5 Mb
Numer of contigs	93	45
GC content (%)	56	57.5
Contig N50	226.4 kb	188.4 kb
Contig L50	9	7
Genes	6222	4399
Protein-coding	5857	4231
Number of RNAs (tRNAs, rRNAs, others RNAs)	54	52

**Table 3 biology-14-01024-t003:** The comparison of genetic traits of KW56^T^ and 2063^T^ strains with other sequenced *Phyllobacterium* genomes.

Strain	dDDH (d4, in %)	C.I. (d4, in %)	ANIb (%)	Aligned (%)	Aligned (bp)
	KW56				
*P. bourgognense* 31-25a^T^	48.1	53.8–61.0	91.09	68.66	4,368,116
*P. brassicacearum* 29-15	23.3	21.0–25.7	79.05	54.96	3,496,502
*P. brassicacearum* STM 196^T^	23.3	21.0–25.7	79.06	55.04	3,502,100
*P. endophyticum* PEPV 15^T^	25.1	22.7–27.5	80.89	59.67	3,796,325
*P. leguminum* ORS 1419^T^	20.5	18.3–22.9	71.72	25.48	1,621,165
*P. pellucidum* BT 25^T^	20.7	18.5–23.1	76.58	51.76	3,292,876
*P. phragmitis* 1N-3^T^	19.7	17.5–22.1	72.10	33.47	2,129,418
*P. myrsinacearum* DSM 5893^T^	20.7	18.5–23.2	76.22	48.26	3,070,563
*P. salinisoli* LLAN 61^T^	20.0	17.8–22.4	71.75	33.60	2,137,656
*P. sophorae* CCBAU 03422^T^	23.1	20.8–25.6	78.91	59.18	3,765,107
*P. trifolii* CECT 7015^T^	58.6	55.8–61.3	93.52	73.69	4,688,571
*P. zundukense* Tri-48^T^	24.3	22.0–26.8	80.07	57.70	3,670,953
2063^T^	21.6	19.3–24.0	77.66	49.23	3,132,251
	2063				
*P. bourgognense* 31-25a^T^	21.7	19.4–24.1	78.15	66.61	3,022,602
*P. brassicacearum* 29-15	28.3	25.9–30.8	84.08	75.22	3,413,166
*P. brassicacearum* STM 196^T^	28.3	25.9–30.8	84.07	75.27	3,415,597
*P. endophyticum* PEPV 15^T^	20.4	18.2–22.8	76.93	63.35	2,874,562
*P. leguminum* ORS 1419^T^	20.9	18.6–23.3	72.41	32.16	1,459,421
*P. pellucidum* BT 25^T^	20.9	18.7–23.3	77.39	66.18	3,002,954
*P. phragmitis* 1N-3^T^	19.6	17.4–22.0	72.66	40.61	1,842,651
*P. myrsinacearum* DSM 5893^T^	20.8	18.5–23.2	77.10	59.92	2,718,871
*P. salinisoli* LLAN 61^T^	19.5	17.3–21.9	72.31	40.71	1,847,234
*P. sophorae* CCBAU 03422^T^	22.9	20.6–25.4	79.62	70.79	3,212,284
*P. trifolii* CECT 7015^T^	21.6	19.3–24.0	78.18	66.57	3,020,677
*P. zundukense* Tri-48^T^	23.8	21.5–26.3	80.81	73.44	3,332,456
KW56^T^	21.6	19.3–24.0	78.17	67.30	3,053,947

**Table 4 biology-14-01024-t004:** Physiological traits of KW56^T^, 2063^T^, *P. brassicacearum* LMG 22836^T^ and *P. trifolii* PETP 02^T^ [5,6,8].

	KW56^T^	2063^T^	*P. br*. *	*P. tr.* *
pH 5	-	-	+	-
4% NaCl	+	+	-	+
37 °C	+	+	-	+
α ketoglutarate	-	+	+	-
arabitol	+	-	-	+
DL-α-amino-n-butyrate	-	-	+	+
L-serine	+	-	+	-
succinate	+	-	+	+
L-aspartate	+	-	+	+
L-alanine	+	-	+	-
propionate	+	+	-	-
D-glucuronate	+	+	-	+
D-galacturonate	+	-	-	+
urease	+	-	+	-
trehalose	-	+	+	+
D-saccharic acid	+	-	-	+
α-hydroxybutyric acid	+	-	+	+
Methyl β-d-glucoside	-	-	+	+
D-gluconic acid	-	-	-	+
D-lactic acid methyl ester	-	-	+	+
beta-ketobutyric acid	+	-	-	+
acetic acid	+	+	-	-
Tween 40	+	-	-	-
methyl pyruvate	-	-	+	+
lactose	-	-	+	-
dextrin	-	-	+	+
enzyme activities				
trypsin	-	+	+	-
α-glucosidase	-	+	-	-

* P. br.—Phyllobacterium brassicacearum LMG 22836^T^; P. tr.—Phyllobacterium trifolii PETP 02^T^.

**Table 5 biology-14-01024-t005:** Fatty acid composition and their relative content (%) in cells of KW56^T^ and 2063^T^ strain.

Retention Time	Fatty Acid	Relative Content (%)	
		KW56	2063
10.73	14:0	1 ± 0.5	0.5
12.89	15:0	tr	0
14.55	16:1	2 ± 1	1.5 ± 0.5
15.00	16:0	13 ± 3	11 ± 1
15.17	3-OH 14:0	5 ± 1.5	4
16.14	17:1	1 ± 0.5	0
16.67	cyclopropyl17:0	2 ± 0.5	tr
16.94	17:0	1	tr
17.07	3-OH 15:0	tr	tr
18.47	18:1	6 ± 2	16 ± 2
18.84	18:0	1	3 ± 1.5
18.95	3-OH 16:0	14 ± 1	9 ± 0.5
19.01	cyclopropyl 19:0	36 ± 3	42 ± 2
19.20	unknown	0	1 ± 0.5
19.92	i19:1	3 ± 0.5	1
20.40	a19:1	4 ± 0.5	3 ± 0.5
20.63	3-OH 17:0	0	1
22.31	3-OH 18:0	1	1
23.40	18:1Me	7 ± 1	6 ± 1
23.76	2-OH 19:1	1 ± 0.5	0
23.91	11-OH 19:0	tr	0
24.15	21:0	3 ± 1	0

## Data Availability

The original contributions presented in this study are included in the article; further inquiries can be directed to the corresponding author.

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
