# Peer review of "Phenotypic, Chemotaxonomic, and Genome-Based Classification of Phyllobacterium Strains: Two Proposed Novel Species, Phyllobacterium chamaecytisi sp. nov. and Phyllobacterium lublinensis sp. nov"

_biology, 2025, doi:10.3390/biology14081024_

Round 1

Reviewer 1 Report

Comments and Suggestions for Authors

Sylwia et al. described the phenotypic, chenotaxomic and genome-based properties of two Phyllobacterium strains isolated from Spanish broom. Based on the comparing with other typical Phyllobacterium strains, the authors found that the two strains show discriminatory characteristics, especially on fatty acid composition. But there raise some questions.

  1. Did the author reinoculated the two strains on Spanish broom and still isolated again?
  2. What is the physiological function of the two strains after inoculated on Spanish broom?
  3. Compare with the strains can form leaf nodules, did the two strains lose leaf nodules formation related genes?
  4. Strain 2063 is closed to P. brassicacearum STM196 and P. brassicacearum 29-15, what is the difference of the host range among these three strains. Same with KW56 with CECT 7015.
  5. The genus name in abstract may be italic.
  6. Table 5, below Retention time, what is the ❡ mean?

Author Response

All responses in attached file

Reviewer 2 Report

Comments and Suggestions for Authors

Comments to the author

The manuscript titled “Phenotypic, chemotaxonomic and genome-based classification of Phyllobacterium strains: Proposal of two novel species Phyllobacterium chamaecytisi sp. nov. and Phyllobacterium lublinensis sp. nov.” thoroughly validated that strains KW56 and 2063 belong to new Phyllobacterium species through phenotypic, chemotaxonomic, and genome-based comparisons with their closest relatives. However, I suggest the following details need improvement:

  1. The entire manuscript should be carefully reviewed to correct formatting errors. For instance, strain names should be italicized, a space should be inserted between numbers and units, and on page 2, line 66, HgCl2 should be corrected to HgCl₂, among others.
  2. In Figure 1, please highlight strains 2063 and KW56 by using bold text or a different color.
  3. On page 3, line 124, please provide a detailed description of the composition or definition of the TY medium.
  4. On page 7, lines 220–227, and page 8, lines 240–245, the results mentioned by the authors lack corresponding figures or tables to support these findings.

Author Response

All responses in attached file

Reviewer 3 Report

Comments and Suggestions for Authors

The authors reported on the phenotypic, chemotaxonomic and genome-based classification of Phyllobacterium strains and proposed the two novel species Phyllobacterium chamaecytisi sp. nov. and Phyllobacterium lublinensis sp. nov.

The intention of this manuscript is not clear because the authors did not meet any of the requirements for the description of bacteria species. However, from the title and all the data presented in the test such descriptions of novel species seems to be the only aim of the entire study. All the experiments have been done and it only needs the appropriate form for such a description. To achieve this the manuscript has to fulfill the minimal standards for valid descriptions of novel bacteria species. See: Parker, C. T., Tindall, B. J., & Garrity, G. M. (Eds.). (2019). International code of nomenclature of prokaryotes: prokaryotic code (2008 revision). International Journal of Systematic and Evolutionary Microbiology, 69(1A), S1-S111. I recommend that the authors take a few descriptions in IJSEM as a guideline for the organization of their descriptions of Phyllobacterium chamaecytisi sp. nov. and Phyllobacterium lublinensis sp. nov.

The fundamental problem of the manuscript is already mirrored in the abstract. It is somewhat chaotic and needs to be rewritten in order to give the reader a chance to understand what this is all about.

For all strains identified as type strains a “T” should be added to the strain designation, e.g. strain 263T.

Please use on the one hand and on the other hand only in combination

The two accession numbers of the draft genome sequences are not valid, please correct this.

Identification of unsaturated FAMEs only by mass spectroscopy is problematic because of potential isomerization of the double bond. For consistent identification of these FAMEs standards are recommended. There are several fatty acid standards available (BacMix, etc.).

There is no formal description of the new species.

Comments on the Quality of English Language

some corrections are needed, see text

Author Response

All responses in attached file

Round 2

Reviewer 3 Report

Comments and Suggestions for Authors

The authors did a very good job in revising their manuscript. However, three issues remain:

- The type strains have to be deposited in two different strain collections from two different countries. CGA is not a strain collection but a depository of genome sequences.

- Mark all type strains with T. This means that KW56 has to be changed to KW56T and 2063 to 2063T throughout the entire text.

- The 16S rRNA gene sequences of the two type strains have to be deposited as well and their accession numbers have to be given in the species description in the protologue.

Author Response

All responses in submitted file
